# Morphological Analysis of the Anatomical Mandibular Lingual Concavity Using Cone Beam Computed Tomography Scans in East Asian Population—A Retrospective Study

**DOI:** 10.3390/diagnostics14161792

**Published:** 2024-08-16

**Authors:** Hee-Ju Ahn, Soo-Hwan Byun, Sung-Ah Che, Sang-Yoon Park, Sang-Min Yi, In-Young Park, Sung-Woon On, Byoung-Eun Yang

**Affiliations:** 1Department of Oral and Maxillofacial Surgery, Hallym University Sacred Heart Hospital, Anyang 14068, Republic of Korea; mathjunior@naver.com (H.-J.A.); purheit@daum.net (S.-H.B.); sungche1219@gmail.com (S.-A.C.); psypjy0112@naver.com (S.-Y.P.); queen21c@hallym.or.kr (S.-M.Y.); 2Department of Artificial Intelligence and Robotics in Dentistry, Graduate School of Clinical Dentistry, Hallym University, Chuncheon 24252, Republic of Korea; denti2875@hallym.or.kr (I.-Y.P.); drummer0908@hanmail.net (S.-W.O.); 3Institute of Clinical Dentistry, Hallym University, Chuncheon 24252, Republic of Korea; 4Dental Artificial Intelligence and Robotics R&D Center, Hallym University Medical Center, Anyang 14068, Republic of Korea; 5Department of Orthodontics, Hallym University Sacred Heart Hospital, Anyang 14068, Republic of Korea; 6Division of Oral and Maxillofacial Surgery, Department of Dentistry, Hallym University Dongtan Sacred Heart Hospital, Hawseong 18450, Republic of Korea

**Keywords:** anatomical limitation, cone-beam computed tomography, dental implant, mandibular lingual concavity

## Abstract

The rising demand for dental implants necessitates addressing anatomical challenges, particularly the shape of the mandible. Incorrectly angling implants can cause lingual perforations, risking damage to the inferior alveolar artery and nerve. This study analyzed 96 cone-beam computed tomography images from individuals aged 20 to 70 (8 males and 8 females) to evaluate mandibular anatomy in four areas: left and right sides and the first and second molars. Mandibular shapes were classified into U, C, and P types. U-shaped mandibles, with a wider crest width, pose the highest risk of lingual perforation. Measurements for U-shaped types included concavity angle, length, and depth. Statistical analyses (T-tests and ANOVA) with a 95% confidence interval showed no significant differences between the left and right sides. However, significant differences based on gender, age, and tooth type were found. The study found U-shapes in 34.6% of cases, P-shapes in 28.9%, and C-shapes in 36.5%, with U-shapes more common in second molars. Understanding these variations enhances the safety and effectiveness of implant procedures and oral surgeries.

## 1. Introduction

In contemporary dentistry, the restoration of missing teeth has been a topic of ongoing discussion. Historically, in the early 20th century, the common approach involved preparing the adjacent teeth for a bridge to replace missing ones. However, by the late twentieth century, there was a notable shift with the introduction of dental implant therapy using titanium metal. Dental implants are widely acknowledged as the preferred method for tooth restoration. Precise treatment planning and a thorough understanding of anatomical structures and their spatial relationships in the relevant area are likely the most effective strategies for minimizing surgical complications [1,2]. 

Lekholm and Zarb (1985) outlined five stages of jaw resorption, ranging from minimal to severe, which were incorporated into implant planning, primarily focusing on the volumetric changes in the alveolar ridge post-edentulism [3]. Their primary focus was on the volumetric changes in the alveolar ridge following tooth loss. As implants gain favor, associated technologies are evolving to address anatomical constraints such as the mandibular lingual nerve and maxillary sinus, including short implants, maxillary sinus elevation, and alveolar bone grafting [4,5]. Despite advancements in these technologies, clinicians often overlook anatomical structures. Recently, a more detailed description of the cross-sectional morphology in the posterior mandibular region has been provided, identifying the presence of lingual concavity and a pronounced slope of the lingual cortex as potential surgical risks [6,7]. However, comprehensive information regarding lingual concavity appears to be limited. Despite the well-documented reliability of osseointegration in dental implants, conducting thorough clinical and radiographic assessments of the ridge at the implant site remains crucial. These evaluations are essential for accurately assessing the dimensions and structure of the alveolar bone, which is critical for precise patient selection, diagnosis, and treatment planning. This necessity is particularly emphasized when considering implant placement in the posterior mandibular region, where an evaluation of the lingual undercut’s position and the inferior alveolar canal is vital for preventing potential surgical complications [8]. Failure to consider such anatomical features or inaccurate implant positioning can lead to lingual perforation, resulting in significant complications, including potential damage to the lingual artery and lingual nerve, as well as infections, pain, and in severe cases, hematoma and airway obstruction (Figure 1) [9,10,11].

To avoid such complications, there is a growing body of research analyzing the types and degrees of mandibular lingual concavity in patients, aiming to provide insights into understanding these anatomical structures and ultimately addressing and mitigating potential complications associated with implant procedures. Moreover, the American Association of Oral and Maxillofacial Surgeons stresses the importance of obtaining pre- and post-implantation computed tomography (CT) imaging to determine appropriate angles, diameters, and positions. Cone-beam computed tomography (CBCT) has been reported as helpful in evaluating the anatomy of the posterior mandible, offering dental imaging with lower radiation exposure compared to conventional CT scans while providing higher accuracy [12]. Implant treatment planning necessitates understanding mandibular anatomical structures and variations across ages, genders, and tooth types. However, current research is predominantly limited to studies focused on America, Europe, or Southeast Asia [13,14,15]. While Watanabe et al. researched the mandible’s morphology in Japanese patients, studies focusing on lingual concavity in East Asia are scarce [16]. Therefore, this study aims to analyze anatomical lingual concavity, specifically within East Asia. The participants in this study are all Korean. Korea is one of the most ethnically homogeneous countries in the world, with over 96% of its population being ethnically Korean. Koreans share genetic markers typical of East Asian populations, yet they possess distinct genetic traits that set them apart from their regional neighbors. They generally have straight, dark hair, brown eyes, and light skin [17].

## 2. Materials and Methods

### 2.1. Patient Selection

The study was conducted according to the guidelines of the Declaration of Helsinki and approved by the Hallym University Sacred Heart Hospital Institutional Review Board (IRB No. 2024-02-014-001). A total of 96 patients meeting the criteria were selected, comprising 48 males and 48 females, evenly distributed across six age groups: 20s (20–29), 30s (30–39), 40s (40–49), 50s (50–59), 60s (60–69), and 70s (70–79). Each age group included 8 males and 8 females. The inclusion criteria were as follows:-Adults aged 19 years or older whose jawbone growth was completed.-Patients who visited Hallym University Sacred Heart Hospital and had their first and second molars present from 2020 to 2023.-Patients without systemic or local conditions, such as osteoporosis or periodontitis, that could alter the condition of the jawbone.-Individuals with no surgical history, such as the removal of benign tumors in the mandible.-Individuals who have not undergone radiation therapy in the oral cavity region.

### 2.2. Measurements

CBCT images were reconstructed into cross-sections using Invivo 6 (Anatomage, Santa Clara, CA, USA). Analysis of the left and right mandibular first and second molars was conducted in 96 patients, resulting in the analysis of 384 cross-sectional images. Landmarks such as the alveolar crest, inferior alveolar nerve canal, and mandibular lingual concavity were used for analysis. Line A was drawn two millimeters (mm) above the superior border of the inferior alveolar nerve. Line A served as the reference for the ridge, and the width of the ridge at the level of Line A was measured as Wb. Additionally, the width of the ridge, which was two mm below the alveolar crest, was measured as Wc. Wb and Wc were used to determine the ridge shape. The height between the baselines of Wb and Wc was measured as Vcb. According to Chan et al.’s criteria, the lingual concavity of the mandible was classified as U, C, or P type. The ridge of the C type was wider than Wb, while the ridge of the U type was wider than Wc. The ridge of the P type had similar widths for Wc and Wb.

The degree of concavity for the U type was separately evaluated. The deepest point of the inner concavity was marked as the A point, and a horizontal line passing through the A point was referred to as Line B. The highest point of the inner concavity was marked as the P point, and the line connecting the A point and P point was termed Line C. The angle between Lines B and C was measured as the concavity angle. The line connecting the A and P points was labeled as Line 1, and its length was measured as the concavity length. The depth of the concavity was determined by drawing a perpendicular line through the deepest point of the concavity, and the length of this perpendicular line represented the depth (Figure 2 and Figure 3) [13].

### 2.3. Statistics

Statistical analysis involved conducting *t*-tests for gender, left–right side, and first and second molars. Analysis of variance (ANOVA) and the Scheffe post-hoc test were performed to determine if there were significant differences across age groups. The confidence level was set at 95%, and the significance level was determined to be 0.05. Additionally, a ratio analysis of lingual concavity types was conducted for the first and second molars, and the results were graphically represented. Statistical analysis was conducted using SPSS IBMv. 27.0 (SPSS, Chicago, IL, USA) software.

## 3. Results

### 3.1. Comparison between Left and Right Sides

No significant differences were observed in the dimensional measurements and the degree of lingual concavity between the left and right sides (Table 1).

### 3.2. Comparison by Tooth Type

Tooth type refers to the first and second molars. In dimensional measurements, the width of the second molar was observed to be greater than that of the first molar. The base width averaged 14.19 mm for the first molar and 15.28 mm for the second molar, while the crestal width was 13.64 mm for the first molar and 15.95 mm for the second molar. Regarding height, the first molar measured 9.09 mm, whereas the second molar measured 8.54 mm, indicating that the second molar is shorter. Overall, the second molar was wider and shorter. Additionally, for the U type, the angle in the second molar was found to be 4.6 degrees smaller than that in the first molar. Significant differences were observed in the dimensional measurements and the angle of lingual concavity degree across the tooth type (Table 2).

### 3.3. Comparison by Gender

Males exhibited greater width and higher height in both the first and second molars than females. Additionally, the angle of the lingual concavity was smaller in males. This indicates that the mandibular bone in males is more developed than in females and that the lingual concavity is deeper in males. Significant differences were observed in both left and right first and second molars in dimensional measurements, specifically in base and crestal widths. Additionally, on both sides of the second molars, there were significant differences in the angle of degree of lingual concavity (Table 3 and Table 4).

### 3.4. Comparison by Age

With increasing age, while other measurements did not show consistent changes, the ridge height steadily decreased. Across different age groups, significant differences were observed in the vertical height of mandibular first and second molars. Additionally, while there were no significant differences in the degree of lingual concavity in the first molars, all three measured items of the degree of lingual concavity showed significant differences in both sides of the second molars (Table 5 and Table 6).

### 3.5. Analysis of Ridge Type Ratio 

In the first molars, the U type represented 19.3%, the P type represented 34.4%, and the C type represented 46.3%. In the second molars, the U type represented 50%, the P type represented 23.4%, and the C type represented 36.6% (Figure 4). Additionally, in the analysis of age distribution according to mandibular ridge types, the U type showed that the 20s age group had the highest proportion, with 27% in both molars. For the C type, the 70s age group had the highest proportion, with 24% in the first molar and 28% in the second molar. In the P type, the 60s age group had the highest proportion in the first molar with 19%, and the 50s age group had the highest proportion in the second molar with 29% (Figure 5).

## 4. Discussion

With the advancement of dental implant technology, implants have become a common choice for restoring missing teeth. Additionally, the improved accuracy and reduced radiation exposure of CBCT compared to medical CT scans have led many dentists to utilize CBCT for diagnostic purposes and virtual surgery planning [18]. Consequently, numerous studies in various dental fields are now utilizing CBCT, and its use is recommended for minor dental surgeries such as implant placement [19]. Our findings show a statistically significant decrease in the vertical height of both sides of the mandible with increasing age (Table 5 and Table 6), which is consistent with many previous studies in the field. These studies have noted considerable variation in the height of the alveolar crest due to potential atrophic changes in the ridge after tooth extraction. Bone remodeling, an ongoing process, reduces ridge height as individuals age, explaining the observed decline in dimensional measurements as age increases [16,20]. However, while previous studies have reported a decrease in mandibular height with time or increasing age post-extraction, our study also demonstrates a decrease in mandibular height with age in deciduous teeth [16].

Furthermore, our study found a wider width, lower mandibular vertical height, and smaller lingual concavity angle in the second molars compared to the first molars (Table 2). The smaller lingual concavity angle indicates a deeper degree of undercut in the concavity [21], which aligns with the results documented by Rajput et al. and Seval et al., who observed that the depth of the submandibular gland fossa was larger at the second molar compared to the first molar [22,23]. Additionally, males exhibit greater mandibular width and vertical height values than females. This is likely primarily due to males being generally larger and having more robust mandibles due to stronger muscle attachments. Secondary factors may include accelerated bone resorption in females following menopause, attributed to estrogen deficiency [21]. Moreover, the relatively smaller mandibles in females may lead to more significant surgical complications. Rajput et al. also observed that males display a more pronounced mandibular lingual concavity in the second molar compared to females, which corresponds with the outcomes of our investigation (Table 3 and Table 4) [23]. This difference could be due to the wider ridge and increased bone thickness observed in males, leading to deeper lingual concavities.

In contrast to the second molars, the absence of a difference in lingual concavity angle among the first molars is attributed to the lesser prevalence of the U type in the first molars. The proportion of the U type in the second molars was 50% (96 out of 192), whereas it was lower, at 19.3% (27 out of 192), in the first molars (Figure 4). The proportion of this ridge anatomy type differs from findings in previous studies. In research conducted in America, the prevalence of type U ranged from 60% to 66%, while in Europe, it was 68%, and in Southeast Asia, it was 27% for the first molars and 38.7% for the second molars [13,15,24,25]. In this study, the proportion of type U in the first molars was 19.3%, while in the second molars, it reached 50%. This suggests that ethnicity may be a variable in mandibular ridge anatomy.

Not only was the proportion of undercut types analyzed per tooth but age group comparisons for each undercut type were also conducted. The first and second molars in individuals in their 20s exhibited over 25% U type. As age increased, the proportion of U type decreased, while the proportions of C or P types increased (Figure 5). This can be attributed to structural changes in the mandible with age. The study also observed decreased ridge height in the mandible across different age groups (Table 5 and Table 6). The mandible begins to resorb from the crestal and buccal aspects, leading to a shift from crestal width to basal width over time. Additionally, the analysis of lingual concavity shapes across age groups, including post-hoc analysis, revealed differences between individuals in their 60s and 70s compared to other age groups (Table 5 and Table 6). This corresponds with other studies showing a similar decrease in cortical bone width in both men and women, starting in their 50s [26].

In this study, there are several limitations that necessitate complementary research. Firstly, this study conducted an anatomical analysis of lingual concavity in the East Asian region. While the measurements of mandibular width, height, and concavity did not differ significantly from previous studies, there were some discrepancies in the proportions of anatomical types. Therefore, further multi-institutional studies are suggested to confirm racial or national differences in ridge types. Secondly, recent trends suggest a move towards a more detailed investigation of concavity angles in undercut-type anatomical ridges [14].

Consequently, subsequent studies are needed to gather samples from a larger number of undercut-type patients for morphological analysis. Lastly, the patient information collected in this study considered the presence or absence of oral surgery experiences, such as radiation therapy, tumors, or cysts, rather than overall systemic health conditions. Therefore, recruiting a patient group with standardized systemic health conditions could facilitate further research. Despite these limitations, this study is significant for its anatomical analysis of mandibular lingual concavity in the East Asian population. Although additional multicenter studies are needed, the fact that the U-type, which has a higher risk of lingual perforation during implant placement in the mandibular second molar, is prevalent suggests that the difficulty of implanting in the mandibular second molar may be increased. Therefore, to ensure safe procedures and minimize complications, it is important to verify the occurrence of lingual perforation at each stage of surgery through careful monitoring for bleeding and palpation.

## Figures and Tables

**Figure 1 diagnostics-14-01792-f001:**
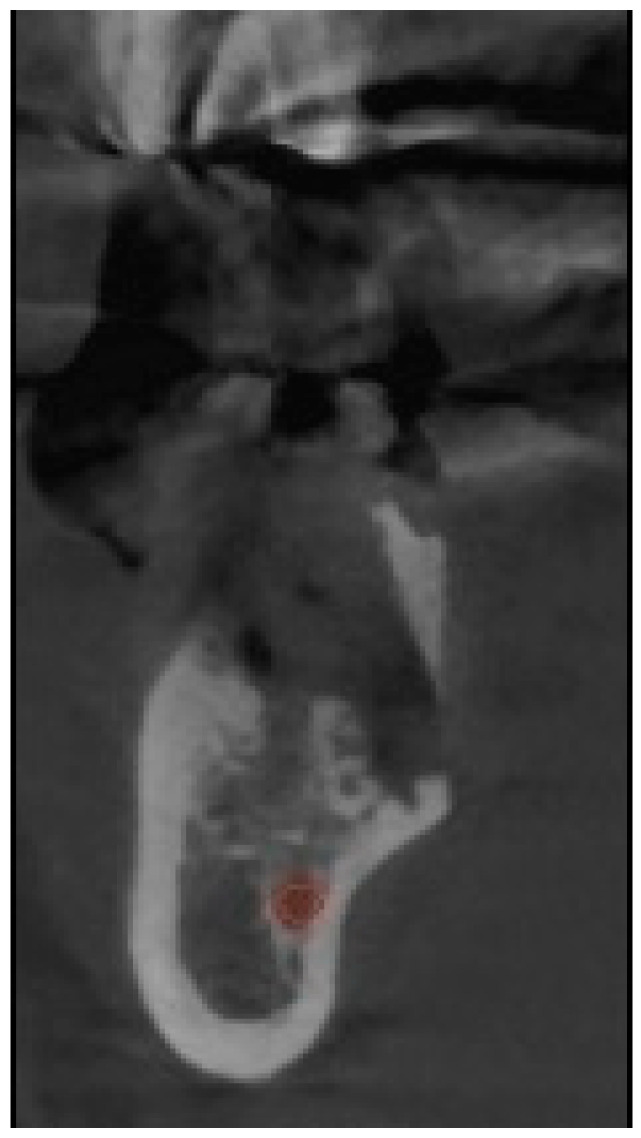
Mandibular lingual perforation resulting from excessive angulation of the implant.

**Figure 2 diagnostics-14-01792-f002:**
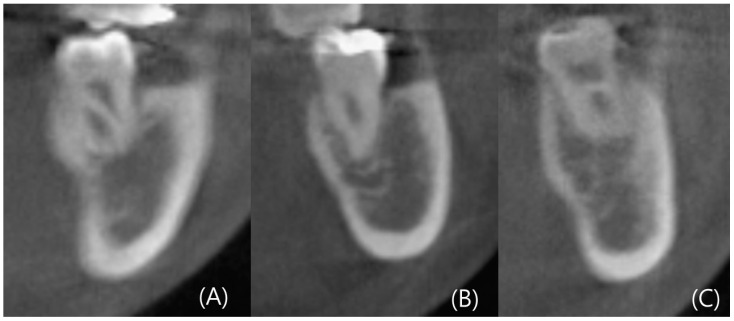
Type of anatomical mandibular ridge (**A**) type U (**B**) type P (**C**) type C.

**Figure 3 diagnostics-14-01792-f003:**
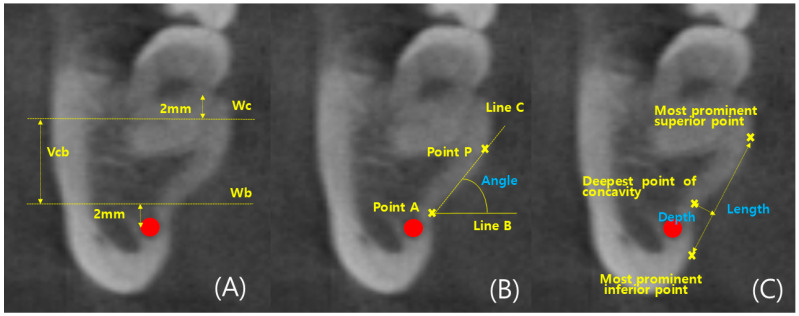
Demonstration of dimensional measurements of alveolar ridges. The red point is the inferior alveolar nerve (IAN) canal. (**A**) Dimensional measurement (**B**) Degree of lingual concavity—Angle (**C**) Degree of lingual concavity—Length, Depth.

**Figure 4 diagnostics-14-01792-f004:**
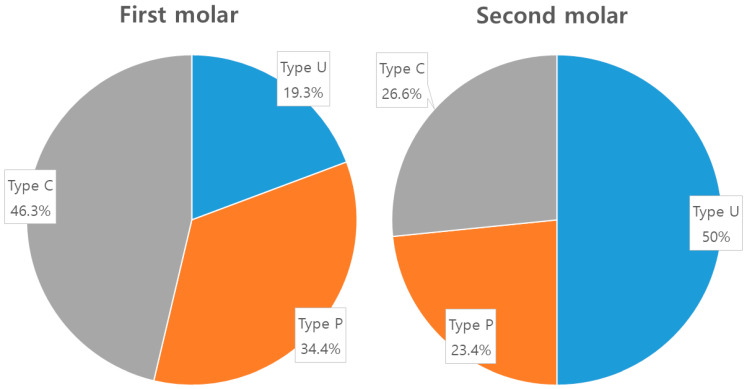
The proportion of anatomical mandibular ridge types in the first and second molars.

**Figure 5 diagnostics-14-01792-f005:**
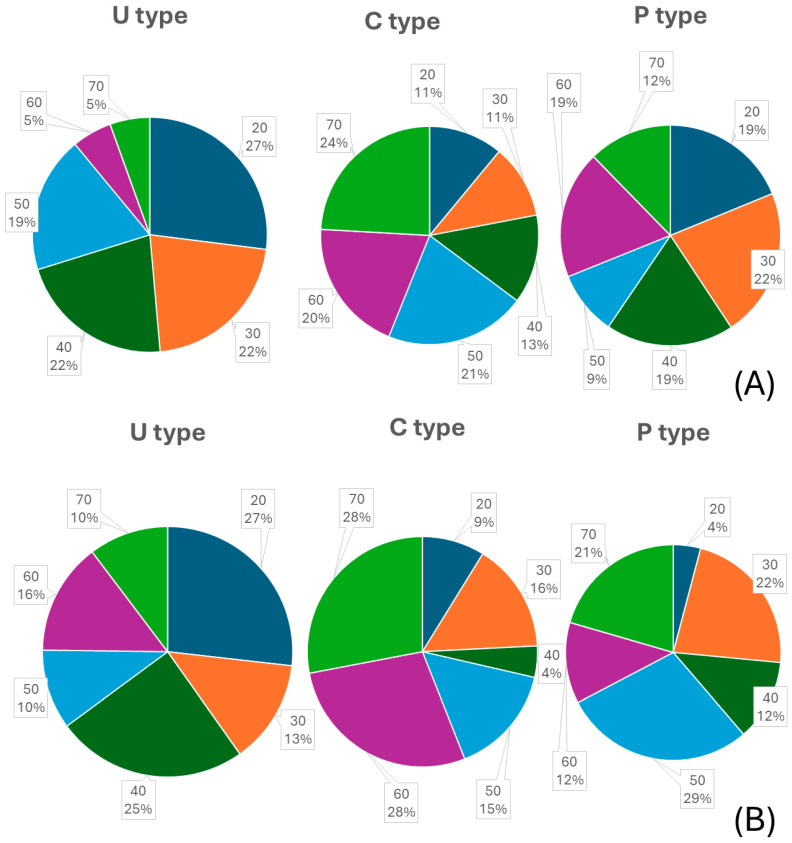
Analysis of age group proportions according to mandibular ridge types. (**A**) First molar, (**B**) second molar.

**Table 1 diagnostics-14-01792-t001:** Statistical analysis of dimensional measurements and degree of lingual concavity on the left and right sides. (SD: standard deviation, unit: mm).

	Tooth Type	Dimensional Measurements			Degree of Lingual Concavity		
		Base Width	Crest Width	Ridge Height	Angle	Length	Depth
**First molar**	Left [SD]	14.25 [2.33]	13.66 [2.14]	9.06 [1.61]	60.2 [7.8]	9.0 [1.6]	1.8 [0.6]
	Right [SD]	14.13 [2.23]	13.62 [2.24]	9.11 [1.65]	61.1 [7.9]	9.0 [2.0]	2.1 [0.6]
	*p*	0.71	0.89	0.97	0.72	0.99	0.24
**Second molar**	Left [SD]	15.13 [2.46]	15.79 [2.47]	8.56 [1.65]	56.1 [6.3]	9.3 [1.9]	2.1 [1.6]
	Right [SD]	15.43 [2.87]	16.11 [2.63]	8.51 [1.55]	56.1 [6.0]	9.3 [1.8]	2.2 [1.9]
	*p*	0.431	0.402	0.824	0.993	0.992	0.171

**Table 2 diagnostics-14-01792-t002:** Statistical analysis of dimensional measurements and degree of lingual concavity according to the first and second molars. (* *p* < 0.05) (SD: standard deviation, unit: mm).

Tooth		Dimensional Measurements			Degree of Lingual Concavity		
		Base Width	Crest Width	Ridge Height	Angle	Length	Depth
**First molar**	Mean [SD]	14.19 [2.27]	13.64 [2.18]	9.09 [1.62]	60.7 [7.7]	9.0 [1.8]	2.0 [1.6]
**Second molar**	Mean [SD]	15.28 [2.67]	15.95 [2.54]	8.54 [1.59]	56.1 [6.1]	9.3 [1.8]	2.1 [1.8]
**All**	Mean [SD]	14.74 [2.47]	14.80 [2.21]	8.82 [1.77]	58.4 [10.4]	9.2 [1.8]	2.0 [1.9]
	*p*	0.004 *	0.003 *	0.003 *	0.002 *	0.507	0.062

**Table 3 diagnostics-14-01792-t003:** Statistical analysis of dimensional measurements and degree of lingual concavity according to gender in the left first and second molars. (* *p* < 0.05) (SD: standard deviation, unit: mm).

Sex	Left First Molar							Left Second Molar					
	Dimensional Measurements				Degree of Lingual Concavity			Dimensional Measurements			Degree of Lingual Concavity		
	Base Width		Crest Width	Ridge Height	Angle	Length	Depth	Base Width	Crest Width	Ridge Height	Angle	Length	Depth
**Male**	Mean [SD]	13.89 [2.23]	14.23 [1.92]	9.21 [1.71]	59.8 [5.1]	9.3 [1.8]	1.8 [0.4]	15.69 [2.45]	16.13 [2.21]	8.89 [1.79]	54.3 [6.0]	9.7 [2.1]	2.1 [1.6]
**Female**	Mean [SD]	13.62 [2.27]	13.09 [2.21]	8.92 [1.50]	60.6 [9.9]	8.8 [1.5]	1.9 [0.7]	14.57 [2.36]	15.46 [2.68]	8.23 [1.43]	57.3 [6.2]	9.0 [1.6]	2.1 [1.8]
**All**	Mean [SD]	13.26 [3.12]	13.66 [3.01]	9.07 [1.61]	60.2 [7.8]	9.0 [1.6]	1.8 [0.6]	15.13 [2.88]	15.80 [2.64]	8.56 [1.53]	56.1 [6.3]	9.3 [1.8]	2.2 [1.9]
	*p*	0.006 *	0.008 *	0.853	0.852	0.541	0.878	0.022 *	0.043 *	0.179	0.043 *	0.216	0.743

**Table 4 diagnostics-14-01792-t004:** Statistical analysis of dimensional measurements and degree of lingual concavity according to gender in the right first and second molars. (* *p* < 0.05) (SD: standard deviation, unit: mm).

Sex		Right First Molar						Right Second Molar					
		Dimensional Measurements			Degree of Lingual Concavity			Dimensional Measurements			Degree of Lingual Concavity		
		Base Width	Crest Width	Ridge Height	Angle	Length	Depth	Base Width	Crest Width	Ridge Height	Angle	Length	Depth
**Male**	Mean [SD]	14.65 [2.14]	14.26 [2.12]	9.29 [1.76]	60.0 [6.9]	9.5 [1.3]	2.2 [0.6]	16.21 [2.96]	16.59 [2.23]	8.64 [1.65]	53.9 [5.4]	9.8 [2.2]	2.2 [2.0]
**Female**	Mean [SD]	13.60 [2.23]	12.97 [2.22]	8.93 [1.53]	62.1 [8.8]	8.7 [2.4]	2.0 [0.7]	14.65 [2.69]	15.62 [2.87]	8.38 [1.45]	57.5 [6.1]	8.9 [1.4]	2.2 [1.9]
**All**	Mean [SD]	14.13 [2.20]	13.62 [2.16]	9.11 [1.66]	61.1 [7.9]	9.0 [2.0]	2.1 [0.6]	15.43 [2.82]	16.11 [2.55]	8.51 [1.55]	56.1 [6.0]	9.3 [1.8]	2.2 [1.9]
	*p*	0.022 *	0.003 *	0.302	0.568	0.403	0.494	0.011 *	0.073	0.396	0.023 *	0.152	0.298

**Table 5 diagnostics-14-01792-t005:** Statistical analysis of dimensional measurements and degree of lingual concavity of the left first and second molars by age. (* *p* < 0.05). (SD: standard deviation, unit: mm, NA : not available).

Age	N		Left First Molar						Left Second Molar					
			Dimensional Measurements			Degree of Lingual Concavity			Dimensional Measurements			Degree of Lingual Concavity		
			Base Width	Crest Width	Ridge Height	Angle	Length	Depth	Base Width	Crest Width	Ridge Height	Angle	Length	Depth
**20~29 ^a^**	16	Mean [SD]	13.67 [2.04]	13.89 [1.61]	10.76 [1.61]	57.5 [11.7]	9.9 [2.4]	2.2 [0.7]	14.06 [1.70]	16.38 [2.20]	10.44 [1.73]	54.5 [7.3]	10.9 [1.6]	2.3 [0.3]
**30~39 ^b^**	16	Mean [SD]	14.08 [3.14]	14.02 [2.37]	9.24 [1.48]	66.5 [4.1]	9.3 [0.8]	1.5 [0.6]	15.83 [3.04]	16.28 [2.86]	8.48 [1.22]	59.0 [4.2]	8.8 [1.3]	2.0 [0.5]
**40~49 ^c^**	16	Mean [SD]	13.91 [1.66]	13.78 [1.54]	8.88 [1.24]	59.7 [5.6]	8.7 [0.6]	8.7 [1.3]	15.18 [2.36]	16.66 [1.98]	8.55 [1.29]	57.4 [7.5]	8.8 [1.0]	1.9 [0.5]
**50~59 ^d^**	16	Mean [SD]	14.26 [2.56]	13.49 [2.38]	8.79 [1.15]	57.9 [5.5]	8.7 [1.3]	1.9 [0.3]	15.34 [2.81]	15.64 [2.56]	8.17 [1.31]	56.8 [4.9]	10.7 [1.7]	2.6 [0.3]
**60~69 ^e^**	16	Mean [SD]	15.52 [1.67]	14.31 [2.25]	8.13 [1.43]	57.6 [NA]	6.2 [NA]	1.1 [NA]	15.46 [1.59]	15.3 [1.54]	7.61 [1.49]	52.5 [4.1]	7.5 [1.5]	1.7 [0.3]
**70~79 ^f^**	16	Mean [SD]	14.08 [2.44]	13.57 [2.39]	7.59 [1.54]	NA	NA	NA	14.92 [2.89]	14.51 [3.04]	7.13 [1.40]	57.6 [5.0]	7.9 [1.6]	2.1 [0.2]
**All**	96	*p*	0.283	0.281	0.002 *	0.373	0.642	0.434	0.443	0.132	0.001 *	0.043 *	0.002 *	0.043 *
		Scheffe post-hoc test			e,f < a,b,c,d						e,f < a,b,c,d	e,f < a,b,c,d	e,f < a,b,c,d	e,f < a,b,c,d

**Table 6 diagnostics-14-01792-t006:** Statistical analysis of dimensional measurements and degree of lingual concavity of the right first and second molars by age. (* *p* < 0.05). (SD: standard deviation, unit: mm, NA : not available).

Age	N		Right First Molar						Right Second Molar					
			Dimensional Measurements			Degree of Lingual Concavity			Dimensional Measurements			Degree of Lingual Concavity		
			Base Width	Crest Width	Ridge Height	Angle	Length	Depth	Base Width	Crest Width	Ridge Height	Angle	Length	Depth
**20~29 ^a^**	16	Mean [SD]	13.65 [1.90]	13.68 [1.56]	10.91 [1.67]	53.0 [3.6]	10.2 [2.5]	2.3 [0.9]	14.01 [2.25]	16.55 [2.22]	9.95 [1.84]	53.6 [5.3]	10.6 [1.6]	2.9 [0.5]
**30~39 ^b^**	16	Mean [SD]	14.06 [2.89]	14.06 [1.49]	8.89 [1.90]	69.2 [3.8]	8.3 [2.6]	1.7 [0.7]	15.66 [3.07]	16.22 [3.07]	8.98 [1.21]	59.8 [2.3]	9.1 [1.3]	2.2 [0.2]
**40~49 ^c^**	16	Mean [SD]	13.67 [1.45]	13.76 [1.55]	9.32 [1.17]	61.5 [5.8]	8.0 [1.4]	2.0 [0.5]	15.30 [2.61]	17.01 [2.46]	8.84 [1.02]	55.7 [4.6]	8.5 [1.5]	2.1 [0.7]
**50~59 ^d^**	16	Mean [SD]	14.09 [2.73]	13.41 [2.41]	8.79 [1.11]	60.3 [5.0]	9.0 [0.9]	2.2 [0.3]	15.48 [3.02]	15.71 [2.75]	8.69 [1.35]	56.6 [6.0]	10.1 [2.2]	2.2 [0.1]
**60~69 ^e^**	16	Mean [SD]	15.13 [1.57]	14.21 [2.42]	8.41 [1.14]	52.8 [NA]	8.3 [NA]	1.6 [NA]	15.52 [1.67]	14.31 [2.25]	8.13 [1.43]	52.0 [1.7]	7.8 [1.6]	1.9 [0.4]
**70~79 ^f^**	16	Mean [SD]	14.14 [2.46]	12.56 [2.68]	8.36 [1.49]	65.8 [1.1]	10.2 [2.7]	2.4 [0.2]	14.08 [2.44]	12.57 [2.39]	7.59 [1.54]	55.4 [4.0]	9.0 [1.2]	1.9 [0.2]
**All**	*96*	*p*	0.143	0.372	0.001 *	0.138	0.583	0.624	0.341	0.492	0.001 *	0.043 *	0.005 *	0.011 *
		Scheffe post-hoc test			e,f < a,b,c,d						e,f < a,b,c,d	e,f < a,b,c,d	e,f < a,b,c,d	e,f < a,b,c,d

## Data Availability

The data supporting this study’s findings are available from the corresponding author upon reasonable request.

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
