# Peer review of "Morphological Analysis of the Anatomical Mandibular Lingual Concavity Using Cone Beam Computed Tomography Scans in East Asian Population—A Retrospective Study"

_diagnostics, 2024, doi:10.3390/diagnostics14161792_

Round 1

Reviewer 1 Report

Comments and Suggestions for Authors

In the Manuscript titled „Morphological Analysis of the Anatomical Mandibular Lingual 2 Concavity using Cone Beam Computed Tomography Scans” the authors evaluated CBCTs of mandibles of 96 patients. They focused on the region of the first and second mandibular molar. They obtained the parameter age, gender / sex, and measurements of the mandible. They found several differences in shape and type.

Queries

-          Do I understand correctly that it is a retrospective radiological study? If that is the case please state it somewhere

-          You used the t-test. Did you do a test for normality, if yes, please state which one and what was the result if no, why not.

-          In table 2 some p-values just show 0 and over all tables the number of digits is not the same. Sine the significant level starts at the second digit after the dot, please report all p-values to the third digit after the dot.

-          I guess in the heading of table 4 is a typo not the left molar but the right molar.

-          You mentioned significant differences between age groups between which age groups are the differences?

-          On page 7 line 158 the maxillary molars are mentioned this confuses me since the study is about the mandible and I think I read somewhere else also about the maxilla please make sure that it is not used in a confusing way.

-          I haven’t seen that you analyzed the age according to the mandibular type, I think that could also be an interesting result.

-          Please add a column in the age tables with the number of n

-          Just a recommendation Figure 4 would be an ideal situation for a pie chart

-          On page 9 line 193ff you talk about larger measurements in males and explain it with menopause. I think the main factor is probably that males are just larger on average and have more robust mandibles because of stronger muscle attachments.

-          Please add limitations of the study

-          Please add a clinical recommendation if possible based on your findings

Reviewer 2 Report

Comments and Suggestions for Authors

The authors conducted an interesting analysis on anatomical characteristics of the mandibula in East Asian population. The study is well designed, however, I have some questions and issues that I would like to address:

Title: Please add "in East Asian population" to the title since this is the "unique" feature of this study.

Introduction:

p1. l. 38-39: "Historically, in the early 20th century, the common approach involved removing adjacent teeth and using bridges to replace missing ones." - really? I think the authors meant to say that a common approach was to prepare the adjacent teeth to retain a bridge and replace the missing tooth. I doubt that missing teeth were replaced by removing more teeth...Please revise.

Materials and Methods:

p.3 l. 97-98: Why was dental implant placement an inclusion criteria for the study population in this present study? The main focus was to analyze the anatomy of the manibula in the molar region. What does implant placement have to do with the participants in this study?

Inclusion criteria:

- Were periodontal disease and degenerative bone diseases (e.g. osteoporosis) exlusion criteria? their presence may severely affect the bone thickness etc..

- please include the ethnical background of your participants since this was an essential feature of your study population.

p.3 l. 108: are the authors certain about Line A being 2 mm above the inferior border of the inferior alveolar nerve? Figure A suggests that superior border was used as a reference. Please clarify.

p.4 l. 116: Please do not use the term "severity" for describing the concavity. Perhaps "extent" or "degree" is a better term.

p.4 l. 116-124: Was this method based on previous work? If so, please include the appropriate references.

Results: Generally, please describe the key results in each section. Although the tables are self explanatory, it would be helpful to the reader to briefly mention where the main differences occurred instead of merely stating that there were signficant differences.

I am not too happy with the presentation of the results. The authors should consider displaying the results with graphs for a better overview of the main differences. 

p.8 l.166 Analysis of ridge type ratio: the autors only stratified the different types of ridges according to the first and second molar. Why not include age and gender in a separate graph? This would be very interesting to the reader!

Discussion: 

p. 9 l. 184-187: I don't quite understand the intention of this phrase. What exactly is the "difference in findings" compared to the other studies? Both studies are reporting a decrease in mandibular height with age. Also, the references to the studies are missing. Please rephrase and add references.

p. 9 l. 209 and 215: Please do not use the term race or nationality. The correct term is "ethnicity". 

Comments on the Quality of English Language

Minor improvements necessary.

Round 2

Reviewer 2 Report

Comments and Suggestions for Authors

All points have been addressed.